# Amplified warming of North American cold extremes linked to human-induced changes in temperature variability

**Russell Blackport** [1] ✉ **& John C. Fyfe**[1]

How global warming is impacting winter cold extremes is uncertain. Previous work has found decreasing winter temperature variability over North America which suggests a reduction in frequency and intensity of cold extremes relative to mean changes. However, others argue that cold air outbreaks are becoming more likely because of Arctic-induced changes in atmospheric circulation. Here we show that cold extremes over North America have warmed substantially faster than the winter mean temperature since 1980. This amplified warming is linked to both decreasing variance and changes in higher moments of the temperature distributions. Climate model simulations with historical forcings robustly capture the observed trends in extremes and variability. A pattern-based detection and attribution analysis shows that the changes in variability are detectable in observations and can be attributed to human influence. Our results highlight that human emissions are warming North American extreme cold temperatures beyond only shifting the winter mean temperature.

Rising global temperatures are expected to decrease the frequency and severity of cold extremes and increase the frequency and severity of hot extremes[1]. However, temperature extremes may change faster or slower than would be expected based on only shifts in the mean if the variability also changes. Several recent high-impact extreme temperature events[2–6] have led to speculation that these extremes are changing beyond simple shifts in the mean. The changes in variability may include changes in variance[7,8], but they could also include changes in higher moments of the temperature distributions, such as changes in skewness and kurtosis[9].

While there is still uncertainty in how temperature variability will change with increasing global temperatures, there is an emerging consensus from models, theory, and observations that daily winter temperature variance over North America decreases. Simulations of both past and projected climate show a very robust decrease in sub-seasonal temperature variance over the northern midlatitudes during winter[7,8,10–12]. The reduction in temperature variance occurs because of the stronger warming of the Arctic relative to lower latitudes, also known as Arctic amplification. This decrease in north-south temperature gradient results in the cold, Arctic air that is displaced southward into the midlatitudes during cold air outbreaks to warm faster than the air that is displaced from the south during warm days[9,10,13]. The decrease in winter variance in the midlatitudes has recently become detectable above background internal variability in observations[8,14,15] and is particularly strong over North America in recent decades[8,11,16].

The reduction in winter temperature variance would strongly suggest that cold extremes are reducing in frequency and intensity even faster than would be expected if there was only a shift in the seasonal mean. This amplified warming of cold extremes over North America is found in climate model projections[17] and observations show a reduction in cold extremes over recent decades over North America[18,19], including amplified warming of extreme cold relative to the mean in observations up until 2014[16]. However, it is not clear how updated observed changes in extremes compare to the mean changes or to model simulations, particularly given recent high-impact cold air outbreaks[4,20]. Less work has been done investigating how higher moments of temperature variability may change, but climate models project an increase in skewness[9,21] and decrease of kurtosis[22] of the daily temperature during winter over the midlatitudes, which would suggest an even stronger reduction in cold extremes relative to the

[1]Canadian Centre for Climate Modelling and Analysis, Environment and Climate Change Canada, Victoria, BC, Canada. ✉e-mail: russell.blackport@ec.gc.ca

mean. Whether these changes in higher moments, and their links to changes in winter temperature extremes have emerged in observations has not been shown.

Despite the established decrease in temperature variance, there have been several high-impact cold-air outbreaks over North America in recent decades[4,20,23]. This has motivated investigation into the potential of Arctic warming and sea ice loss to influence cold extremes[20,24–36]. Several of these studies suggest that changes in the stratospheric polar vortex and/or jet stream caused by Arctic warming have contributed to more frequent and persistent cold extremes over North America[24–27,30,32]. It has also been suggested that climate models may not be capable of capturing these midlatitude circulation responses[37–41] and that cold extremes may increase in frequency[26,32], or at least, not decrease as fast as models predict[32,39,41,42]. If the suggestion that cold extremes are increasing is correct, there would need to be changes in the shape of winter temperature distributions that go beyond changes in variance to reconcile these conclusions with the well-established decrease in temperature variance. The lack of consensus across studies[39,43], and in particular the disagreements between model and observations-based studies[39], leads to lower confidence in projections of cold extremes.

The disagreements between studies motivate us to investigate how winter cold extremes and temperature variability are changing over North America in both observations and models. Instead of investigating only the temperature variance[7,8], we investigate how the full shape of the wintertime temperature distributions are changing over recent decades using quantile regression[16,44,45]. We focus on the tails of the distributions and whether changes in extremes can be characterized by shifts in mean alone, or whether changes in variance or higher moments are involved. To assess model fidelity, we directly compare observed changes to the changes found in large ensemble model simulations with historical forcing. We find that the spatial patterns of the trends in the shape of the temperature distributions are similar between models and observations, so we then apply a pattern-based detection and attribution analysis. Finally, we investigate the role of the sea ice loss using model experiments forced with only a reduction in sea ice.

## Results

### Trends in winter mean and extreme temperatures
We begin by comparing trends in winter (December–January–February; DJF) mean and extreme near-surface temperatures over North America using daily averaged data from ERA5 reanalysis over the 1980–2022 period. For better comparisons to models that have different amounts of global warming, the trends in mean and extremes are divided by the global, annual mean temperature trend so that trends are plotted in units of °C per °C of global warming. The winter mean trends show warming over most of North America, but the trends are very weak over the central part of the continent (Fig. 1a). We examine trends in extremes by using quantile regression[16,44,45] to calculate trends in the 2nd and 98th percentile of the daily mean winter temperatures. The 2nd percentile trends, representing the coldest winter days, differ from the mean trends with strong warming throughout the continent (Fig. 1b). Averaged over the United States and southern Canada (30°–52°N, land only), the coldest days have warmed about 2.2 times more than the winter mean and 3.2 times more than the global, annual mean trend. The 98th percentile trends, representing the warmest winter days, also differ from the mean with weak, but not statistically significant cooling over much of Canada (Fig. 1c), opposite to what is seen in the mean trend.

The differences between the extreme and mean trends show that the 2nd percentile days are warming faster than the mean over most of North America (Fig. 1d) and the 98th percentile days are cooling relative to the mean over most of Canada (Fig. 1e). The faster warming of the coldest days (2nd percentile) extends further south than the slower

warming of the warmest days (98th percentile), suggesting that over the southern part of North America, there could be changes in the temperature distributions beyond variance, which we will return to in the next section. We note that after controlling for the false discovery rate[46], there are few grid points that are statistically significant for either the trends in extremes or the differences between the mean and extremes. However, below we will show that trends in regional averages and patterns are statistically significant. Nearly identical results are found in other reanalysis products and in gridded observational data (Supplementary Fig. 1). Similar results are also found if we use 50th percentile trends instead of the mean (Supplementary Fig. 2).

We next compare the reanalysis trends to modelled trends over the same time period from initial condition, large ensemble experiments from seven models with historical forcing (see the "Methods" section). The multi-model mean trends show many of the same features as the ERA5 trends (Fig. 1f–j). As in ERA5, the 2nd percentile trends are warming faster than the mean over most of North America, and while we do not see cooling of the 98th percentile trends, they are warming slower than the mean over Canada. These results are robust across each of the seven models (the stippling in Fig. 1f–j represents where the models disagree on the sign of the trends in the ensemble means).

Figure 2a compares the magnitude of the ERA5 and modelled trends averaged over the United States and Southern Canada, taking into account the uncertainty from internal variability. The uncertainty in the ERA5 trends is calculated using a bootstrapping approach (see the "Methods" section). Despite the few statistically significant trends at individual grid points, the magnitude of the 2nd percentile trends averaged over the domain in ERA5 is statistically significant and is on the higher end of the ensemble spread in each model. The mean trends are in the middle of the ensemble spread, and the 98th percentile trends are on the lower end of the ensemble spread. The differences between the extreme and mean trends for the average over the United States and Southern Canada are plotted in Fig. 2b. The difference between the 2nd percentile and mean averaged over this region in ERA5 are marginally statistically significant and within the model ensemble spread. Figure 2 also highlights the robustness of the amplified warming of the extreme cold and weaker warming of extreme warm temperatures across the different models. If we examine less extreme trends (6th and 94th percentile), we find weaker but less noisy trends, resulting in similar signal-to-noise ratios as the more extreme trends (Supplementary Fig. 3).

### The changing shape of winter temperature distributions
To quantify how the full shape of the temperature distributions are changing, we calculate the trends for all percentiles from 2nd to 98th with an interval of 2%. The zonal mean over North America of these trends is plotted in Fig. 3. In both ERA5 and the multi-model mean, nearly all latitudes and percentiles show warming, with stronger warming at higher latitudes and the lowest percentiles (coldest days). To highlight the changes in the shape of the temperature distributions, we show trends as a function of percentile with the mean trend removed in Fig. 3b, e. Both ERA5 and the models indicate that north of about 45°N, colder days are warming faster than the mean, and warmer days are warming slower. The approximately antisymmetric trends on the cold and warm side of the distributions represent a decrease in variance, which is confirmed by the variance trends (Supplementary Fig. 4) and by previous studies[8,10,11]. South of about 45°N, it is only the very coldest days that are warming faster than the mean, suggesting that higher moments beyond variance are changing.

Figure 3c, f shows the trends as a function of percentiles after both the mean and variance trends are removed (see the "Methods" section), representing trends linked with the higher moments of the distribution. In ERA5, south of about 45°N, there are strong positive trends of the coldest days (<10th percentile), weak cooling of the

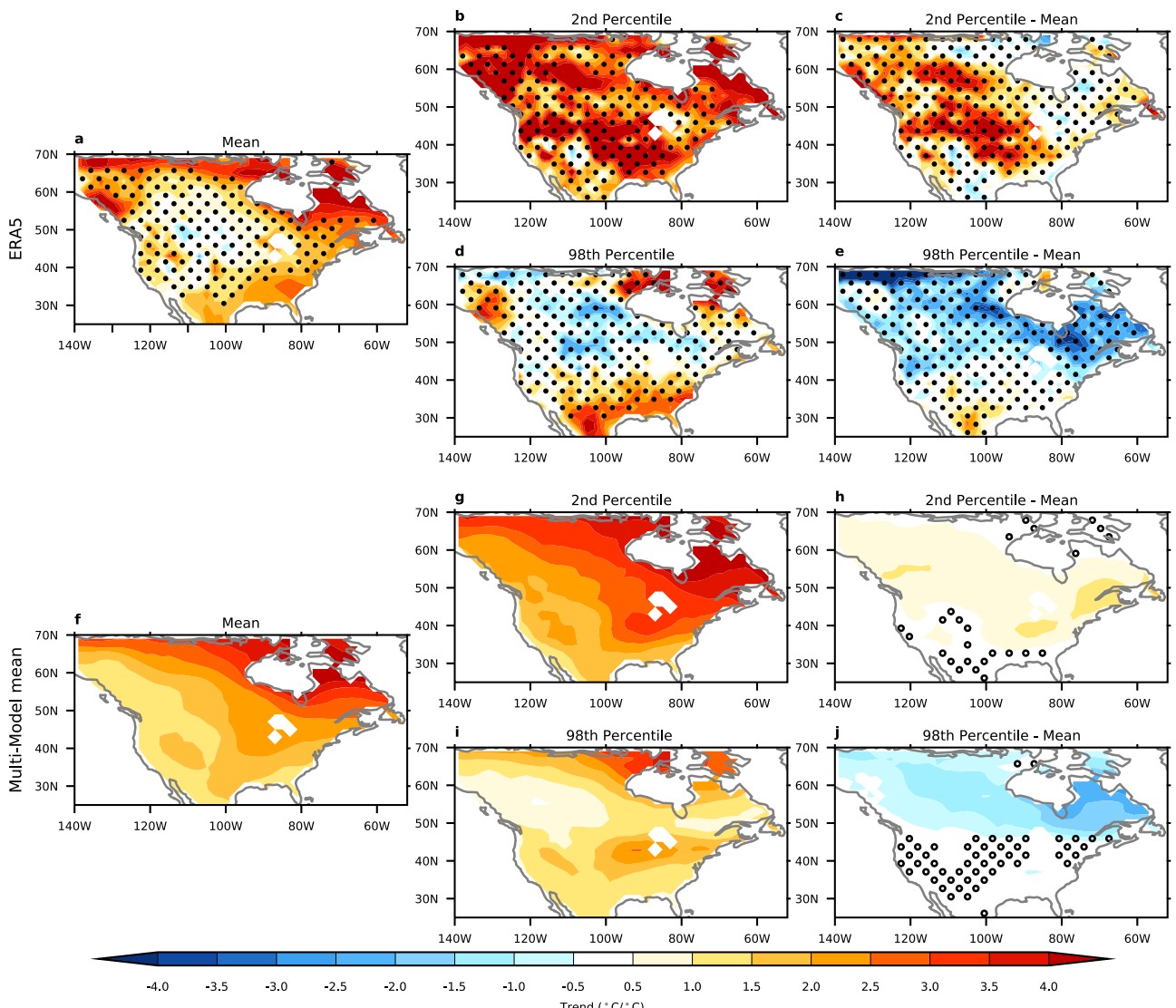

**Fig. 1 | Trends in winter mean and extreme temperatures. a–e** The 1980–2022 winter near-surface temperature trends in ERA5 reanalysis for the mean (**a**), 2nd percentile (**b**), the difference between the 2nd percentile and the mean (**c**), 98th percentile (**d**) and the difference between the 98th percentile and the mean (**e**). **f–j** As in (**a–e**), but for the multi-model mean. All trends are divided by the corresponding global, annual, and mean temperature trends so that the units of the trends are °C per °C of global warming. The stippling in (**a–e**) represents where the ERA5 trends are not statistically significant using a bootstrap approach after controlling for the false discovery rate of 0.1. The stippling in (**f–j**) indicates where more than one of the models disagrees with the sign of the trend in the ensemble mean.

moderately cool days, and weak warming over the warm days that are linked with the higher moments. This pattern is consistent with an increase in skewness, which is confirmed by calculating the trends in skewness directly (Supplementary Fig. 4). The models show similar results that are robust across models, but the warming trends of the coldest days and increasing skewness are only seen in a narrower band around 40°N.

It has been noted that the observed temperature trends over North America show substantial differences between the months within winter, with December and January showing warming and February showing cooling[32]. The trends as a function of the percentile in each individual winter month from ERA5 confirm this (Supplementary Fig. 5). However, even in February, the coldest days are still warming despite the cooling in the winter mean, and the changes in shape of the temperature distributions are similar in all three months.

Returning to the cold extremes, Fig. 4 shows the decomposition of the 2nd percentile trends into trends linked to mean, variance, and higher moments. In ERA5, over the midlatitudes, only about half of the trends in extreme cold are linked to the increase in mean temperature

(Fig. 4a). North of about 45°N, the other half of the trends in extreme cold are linked to the decreases in variance. However, south of 45°N, changes in variance play little role, and the amplified warming is linked to changes in higher moments of the temperature distributions. These differences between north and south of 45°N can also be seen by looking at the changes in temperature distributions from representative grid points (Supplementary Fig. 6). The model decomposition is similar (Fig. 4b), but the mean can explain a more substantial fraction of the extreme cold trends, and the magnitude of the trends linked with variance and higher moments are weaker. We note that the larger contribution from the mean in the model trends does not necessarily represent model error and may be a result of internal variability in the ERA5 trends, which is very large (see, e.g. Fig. 2).

**Detection and attribution**

The similar patterns in observed and model trends motivate us to use pattern-based methods to detect and attribute the observed changes to human activity. We use standard regression methods[47,48] (see the "Methods' section), where we regress the spatial pattern of the

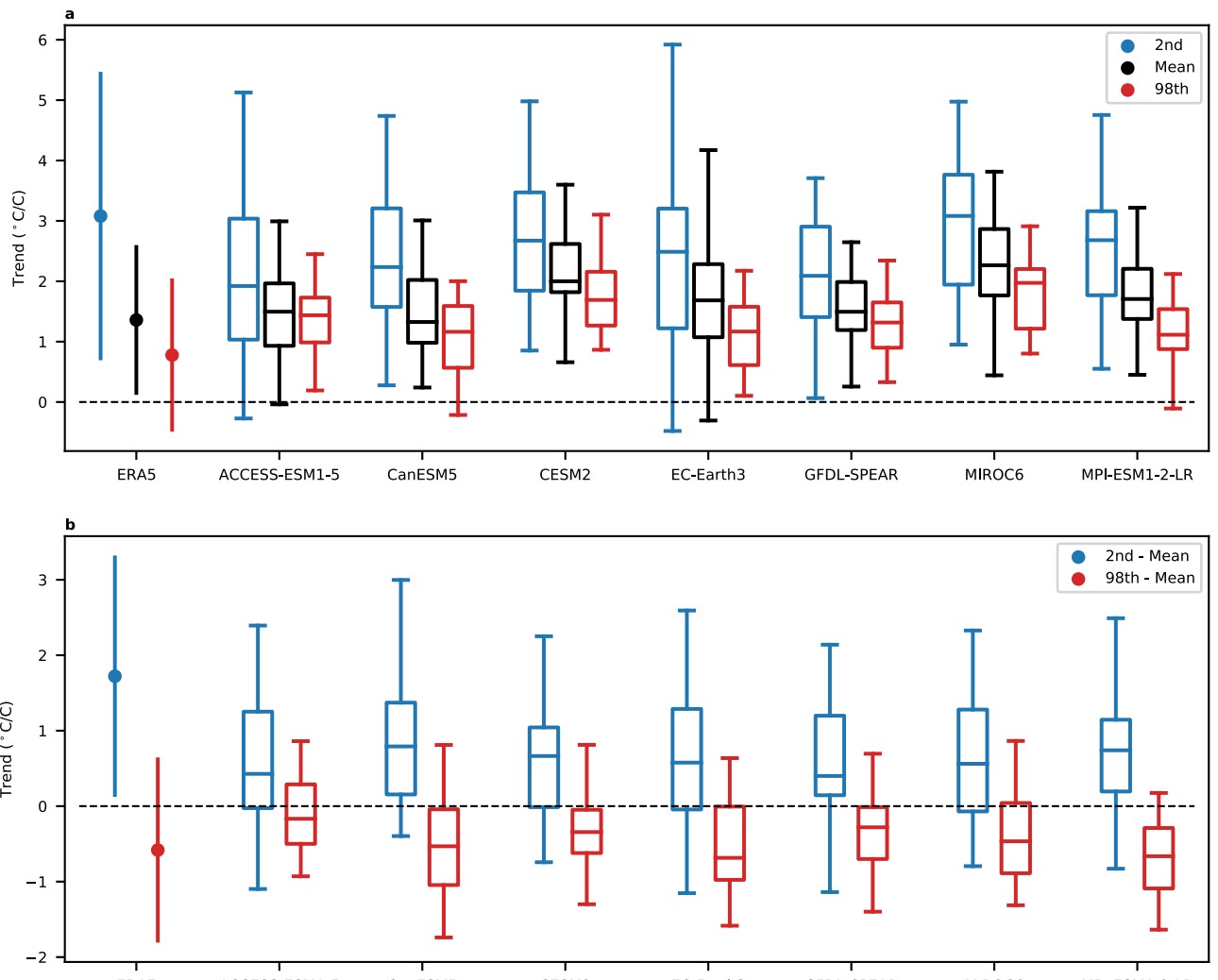

**Fig. 2 | Comparison of trends in reanalysis to the model ensemble spread. a** The magnitude of the 1980–2022 trend in winter 2nd percentile temperature (blue), winter mean temperature (black), and 98th percentile temperature (red) from ERA5 (dots). Trends are averaged between 140°–52°W and 30°–52°N, land only. The error bars for ERA5 represent the 2.5–97.5% range and are calculated using a bootstrapping approach. The box and whisker plots represent the ensemble spread in trends from each of the seven models. The box represents the inner quartile range, the whiskers represent the 2.5–97.5% range of trends, and the line represents the median trend. All trends are divided by the corresponding global, annual, and mean temperature trends so that the units of the trends are °C per °C of global warming. **b** As in (**a**), but for the difference between the 2nd percentile and mean trends (blue) and the difference between the 98th percentile and the mean trend (red).

observed trends from ERA5 onto the model trend patterns (or fingerprints) to calculate regression coefficients or scaling factors. A scaling factor that is statistically above zero indicates that the model pattern is detectable in observations, and a scaling factor consistent with one indicates that the magnitude of the model trend pattern is consistent with observations. Uncertainty in the scaling factor is determined by calculating the regression coefficients from internal variability in the models (see the "Methods" section). We apply these methods to the trends in mean, variance, and skewness (two-dimensional fields as a function of latitude and longitude) and to the trends as a function of percentile, the trends as a function of percentile with the mean trends removed, and the trends as a function of percentile with both the mean and variance trends removed (three-dimensional fields as a function of percentile, latitude, and longitude).

Figure 5 shows the resulting scaling factors using a one-signal analysis with the model patterns from the multi-model mean of the historical simulations. The mean temperature pattern is detectable in observations, albeit by a small margin. The variance trends are highly detectable, with a signal-to-noise ratio substantially higher than the mean trends. The scaling factor for the variance pattern is about 1.7,

which suggests that the model signal is too weak. However, because the uncertainty range overlaps with one, the stronger observed signal could be a result of an unusually strong trend arising from internal variability. The skewness pattern has a scaling factor close to one, with the lower bound of the uncertainty range barely overlapping with zero (7% percent of the distribution is below zero). Thus, it is not quite detectable at the 5% confidence level yet. The trends as a function of percentile are detectable, but interestingly, when the mean trends are removed, the signal-to-noise ratio increases. Even after removing both the mean and variance trends, resulting in changes in the temperature distributions associated with the higher moments of variability, the pattern is still detectable in observations. Three of the large ensemble experiments used here also have corresponding single forcing experiments, which can be used to attribute the signal to greenhouse gasses, anthropogenic aerosols, or natural forcings (see the "Methods" section). We find that that the greenhouse gas scaling factors are detectable and are similar to all historical forcing experiments (Supplementary Fig. 7), indicating the observed trends can be attributed to anthropogenic greenhouse gas forcing.

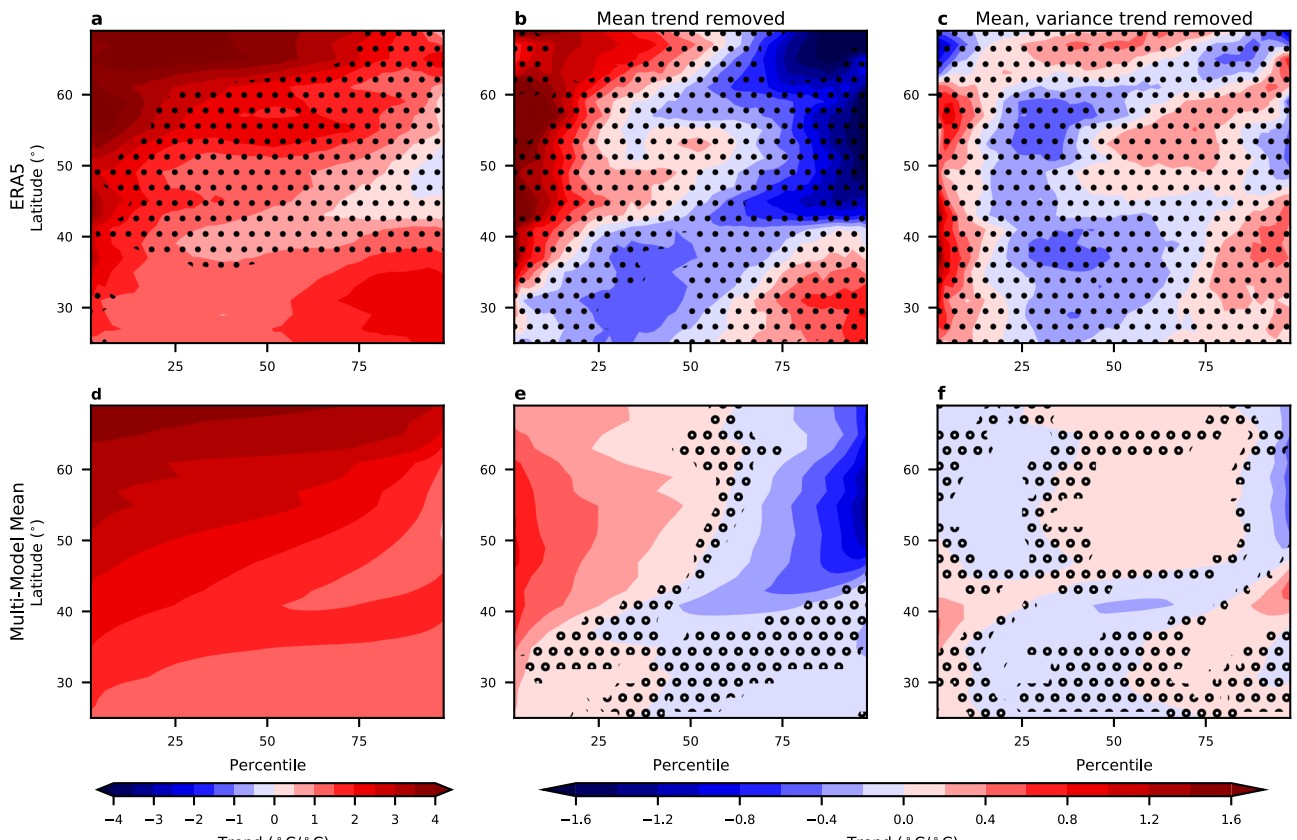

**Fig. 3 | Zonal mean of winter temperature trends as a function of percentile.** **a** Zonal mean (averaged between 140° and 52°W, land only) of winter temperature trends as a function of percentile and latitude for ERA5 over 1980–2022. **b** as in (**a**), but with the winter mean trend subtracted from the trend. **c** As in (**b**), but with both trend in mean and variance subtracted from the trends. **d**–**f** As in (**a**–**c**), but for the multi-model mean. All trends are divided by the corresponding global, annual, and mean temperature trends so that the units of the trends are °C per °C of global warming. The stippling in (**a**–**c**) represents where the ERA5 trends are not statistically significant using a bootstrap approach after controlling for the false discovery rate of 0.1. The stippling in (**d**–**f**) indicates where more than one of the models disagrees with the sign of the trend in the ensemble mean.

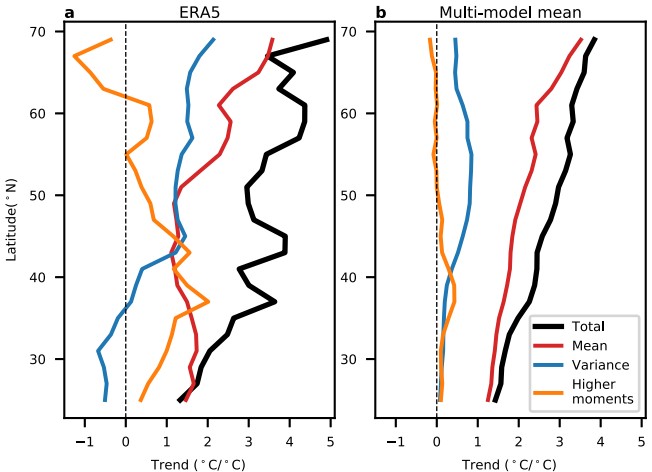

**Fig. 4 | Decomposition of trends in extreme cold temperatures. a** The zonal mean (averaged between 140° and 52°W, land only) of the 2nd percentile winter temperatures over North America (black), and the component that can be explained by the trend in the mean (red), variance (blue), and higher moments (orange). **b** as in (**a**), but for the multi-model mean. All trends are divided by the corresponding global, annual, and mean temperature trends so that the units of the trends are °C per °C of global warming.

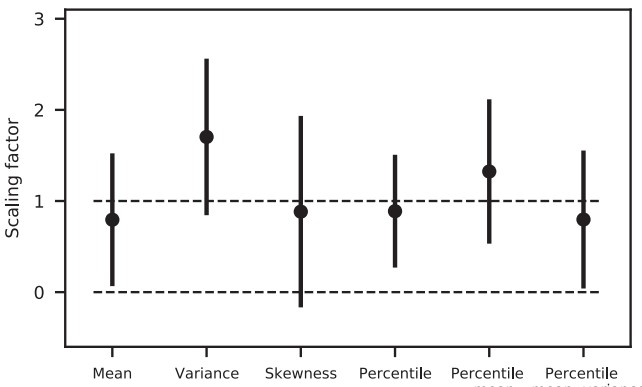

**Fig. 5 | Scaling factors calculated from the detection and attribution analysis.** The scaling factors for the multi-model mean fingerprint of the trend in mean, variance, skewness, trends as a function of percentile, trends as a function of percentile with the mean trend removed, and trends as a function of percentile with both the mean and variance removed. The uncertainty on the scaling factors represents the 5–95th percentile range calculated from internal variability.

## The role of sea ice loss

We next examine the role of sea ice loss in the changes in the shape of the temperature distributions using model experiments that isolate the impact of sea ice loss. Previous work has shown that decreases in winter temperature variance over North America are driven by Arctic amplification or sea ice loss[7,10,11,16,49,50], but the mechanism behind the changes in higher moments is less clear. One study argued that the projected increase in skewness is caused by melting snow cover[21], while another argues that it is caused by the reduction in the meridional temperature gradient, albeit for temperature above the boundary (850 hPa)[9].

To examine the role of sea ice loss, we use simulations from the Polar Amplification Model Intercomparison Project (PAMIP)[51] (see the "Methods" section). Figure 6a–c shows the zonal mean of the temperature as a function of percentile in response to the combined influence of increasing sea surface temperatures and decreases in sea ice concentrations. Despite different experimental designs and different models, the response is qualitatively similar to the historical experiments shown in Fig. 3, with generally enhanced warming of the cold extremes and reduced warming of hot extremes north of 40°N, and some additional changes in the higher moments south of 40°N. We note that the enhanced warming of cold extremes is shifted further to the south relative to what is seen in the historical simulations (Fig. 3). In response to only sea ice loss (Fig. 6d–f), there is strong Arctic amplification and stronger warming of the colder days that are robust across all models. Despite some weak cooling at lower latitudes over most percentiles, the coldest days still warm at all latitudes. Over the midlatitudes, the coldest days warm faster than the mean, and the patterns are similar to the experiments with both sea ice loss and SST warming. The magnitude of the responses relative to the mean is weaker in response to only sea ice loss, which could be due to other processes contributing, but could also be because the Arctic temperature response is likely underestimated in the prescribed sea ice loss experiments[52–54].

These results show that the Arctic warming caused by sea ice loss plays a large role not only in the reduction in temperature variance but also in the changes in higher moments that are linked with faster warming of the cold extremes relative to the mean. Our results are consistent with the previous work arguing that the increase in skewness seen in model projections is a result of changing temperature gradients[9]. Specifically, at these lower latitudes (<45°N), it is only the very coldest days that are influenced by Arctic air. Therefore, it is only these coldest days that warm in response to Arctic warming, and this results in an increase in skewness. Because there is little change in mean temperature in the midlatitudes in response to only sea ice loss, it is unlikely that reduced snow cover plays a role in the sea ice loss experiments, but we cannot rule out that it also contributes to the higher moment changes in the observations and historical simulations.

## Discussion

We have shown that the strong warming of winter cold extremes over North America since 1980 cannot be explained by a shift in the mean temperature or even a change in temperature variance in some regions. Over the more northern regions covering most of Canada, cold extremes are warming faster than the winter mean, consistent with the previously identified observed decrease in temperature variance[7,8,10,11]. However, further south over much of the United States, where there is little change in temperature variance, we also find amplified warming of cold extremes that are linked to changes in higher moments of the temperature distributions, such as skewness. The changes in higher moments suggest that changes in cold extremes cannot necessarily be inferred from changes in mean and variance alone. The amplified warming of cold extremes and changes in variability are robustly captured across all climate models that are forced with historical forcing. Surprisingly, we find that changes in

temperature variability are easier to detect (i.e., they have a higher signal-to-noise ratio) in observations compared to the changes in winter mean temperatures. This is likely because the changes in variability are driven by Arctic amplification, which is extremely robust in observed trends over the past 40 years[55].

Some previous studies have argued that as the Arctic continues to warm, cold extremes over North America may become more common, or at least not decrease substantially[26,30,32,41], in contrast to what is predicted by models. We find no evidence of this in observed or modelled trends of cold extremes or in how the shape of the temperature distributions is changing. We find that cold extremes over North America are warming substantially faster than both the winter mean temperature over North America and the global, annual mean temperature. We also find that observed cold extremes over North America are, on average, warming faster than historical simulations, albeit within the range expected from internal variability. In addition, we find that Arctic warming caused by observed sea ice loss contributes to the warming of cold extremes over North America, in agreement with studies examining the response to projected sea ice loss[49,56].

The reason for the discrepancies with studies claiming cold extremes may become more common is likely that these previous conclusions were based on the observed links between the Arctic and the midlatitudes seen in shorter-term (daily-to-decadal) variability[20,26,27,30,32], not long-term trends of the winter cold extremes. The agreement between models and observations indicates that previously identified discrepancies between model-based observation-based studies[39,41] may have occurred because of misinterpreting observations, not model error. In addition, some of the conclusions have been based on seasonal or monthly mean temperature trends[24,32,39,41], which we show are not necessarily representative of the trends in cold extremes (see, e.g. Fig. 1a, b).

It is important to note that despite our results, winter cold extremes over North America will continue to occur. Winter temperatures over North America in the current climate have the highest variance[57] and are some of the most strongly negatively skewed[9,21] on Earth. This means that extreme deviations below the mean are expected to continue to occur in the future, even with rising global temperatures. However, because of the increasing mean temperatures, combined with the changes in temperature variability, cold extremes over North America will occur less frequently, and when they do occur, they will be less intense.

## Methods

### Reanalysis data

We use daily averages of hourly near-surface temperature from ERA5[58] over the period 1979–2022. We define winter to be the December–January–February (DJF), and the year corresponds to the year of January and February (e.g. December 2021, January 2022, February 2022 is considered 2022). We also examine daily average near-surface temperature data from NCEP-DOE reanalysis 2[59] and JRA-55[60] to test the sensitivity of the trends to the chosen reanalysis product. For the gridded observations, we use daily average temperature data from Berkeley Earth[61].

### Model data

Large ensemble historical simulations from seven different models are used in this study: ACCESS-ESM1-5[62](40), CanESM5[63] (50), CESM2[64,65] (50), EC-Earth3[66] (50), GFDL-SPEAR-MED[67] (30), MIROC6[68] (50), MPI-ESM1-2-LR[69] (30). The number in parentheses indicates the number of ensemble members used for each model, which adds up to 300 ensemble members in total. These are all the models (to our knowledge) from the latest generation of models that have large ensembles with at least 30 members over the 1979–2022 period at the time the analysis was performed. For the CESM2 model, a total of 100 ensemble

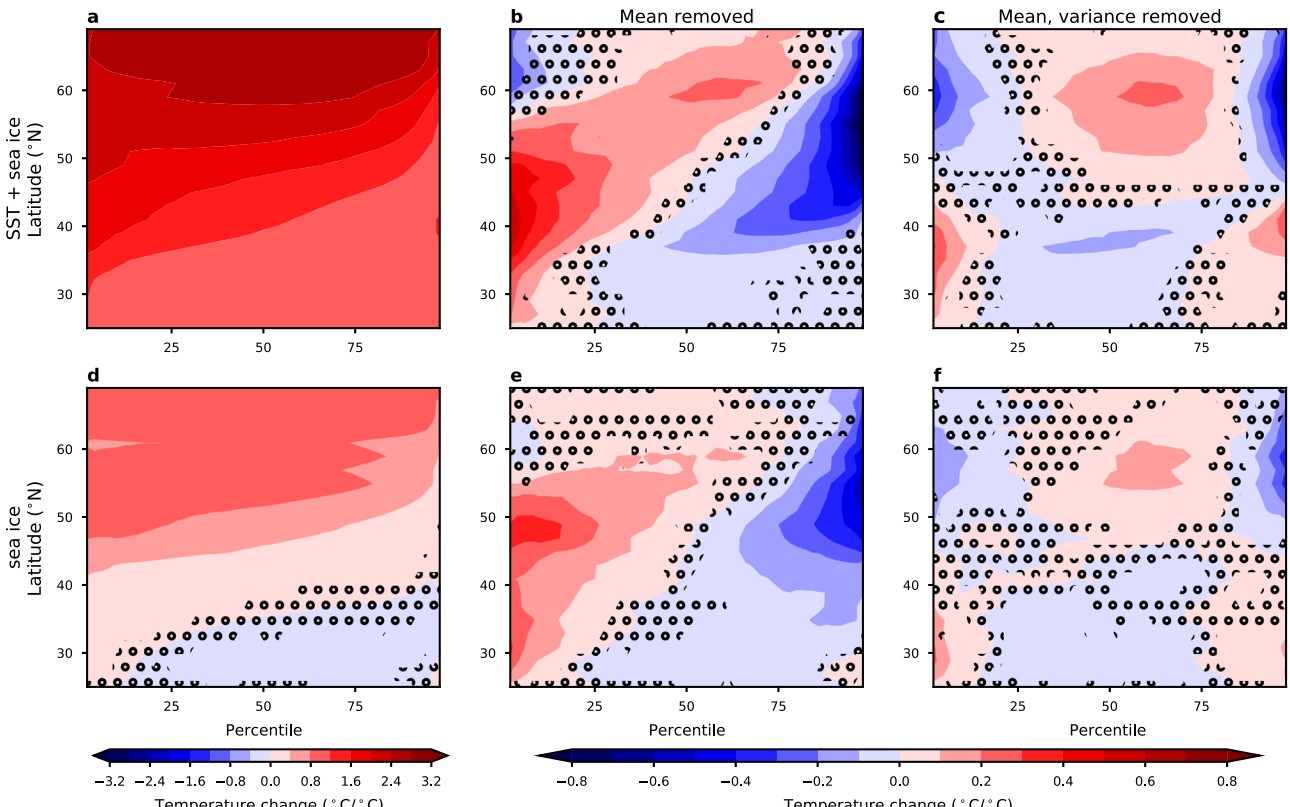

**Fig. 6 | The impact of sea ice loss on winter temperature percentiles. a** Zonal mean (averaged between 140° and 52°W, land only) change in daily winter temperature as a function of percentile and latitude in response to sea surface temperature (SST) warming and sea ice loss from PAMIP simulations. **b** as in (**a**), but with the winter mean change removed. **c** As in (**b**), but with both the mean and variance changes removed. **d**–**f** As in (**a**–**c**), but in response to only sea ice loss. All changes are divided by the corresponding global, annual, and mean temperature change in response to SST warming and sea ice loss, so that the units of the changes are °C per °C of global warming. The stippling indicates where more than one of the models disagrees with the sign of the change in the ensemble mean.

members exist, but 50 of the members used historical biomass burning forcing that is different from that of the Coupled Model Intercomparison Project (CMIP6). To be consistent with the other models, we have only analysed the 50 members that include the forcing from CMIP6. For EC-Earth3, we use ensemble members labelled *r*101–150. The different ensemble members within each model ensemble differ only in their initial conditions. All models are forced with historical forcing until 2014 and then projected forcing thereafter. The projected forcing used is different for each model, depending on data availability. SSP5-8.5 was used in EC-Earth3, GFDL-SPEAR-MED, MIROC6, and MPI-ESM1-2-LR, SSP3-7.0 was used for CESM2 and ACCESS-ESM1-5, and SSP2-4.5 was used in CanESM5. The different scenarios are unlikely to have a large effect on the results because the projected forcings are similar over the period we have analysed (2015–2022). In addition, all trends are divided by the global mean temperature trend.

Three of the models (CanESM5, MIROC6, CESM2) also have single forcing large ensemble runs available. For two of the models (CanESM5, MIROC6), the data are a part of the Detection and Attribution Model Intercomparison Project (DAMIP)[70]. We use only greenhouse gas (GHG), only anthropogenic aerosols (AER) and only natural (volcanic and solar) forcing simulations (NAT). Each ensemble consists of between 10 and 50 ensemble members. The CESM2 simulations[71] are not a part of DAMIP but have nearly identical forcing compared to the other models. The GHG and AER simulations have identical forcing to the DAMIP simulations, however there is no corresponding 'natural only' ensemble from CESM2. Instead, there is an 'Everything Else' (EE) experiment that includes all forcing apart from GHG, AER and biomass burning. This EE experiment primarily included natural forcing but also included land cover changes. We consider the EE experiment as a natural forcing for our analysis.

To investigate the role of sea ice loss we use three experiments from the Polar Amplification Model Intercomparison Project (PAMIP)[51]. Five models are used with the required data for all three experiments: AWI-CM-1-1-MR, CESM2, CanESM5, FGOALS-f3-L, IPSL-CM6A-LR. All experiments are atmosphere-only time slice experiments with prescribed sea surface temperatures (SSTs) and sea ice concentrations (SICs) with year 2000 radiative forcing. The first experiment (piSST-piSIC) is forced with pre-industrial SST and SIC. The second experiment (piSST-pdSIC) is forced with pre-industrial SSTs, but present-day (1979–2008) SIC. The third experiment (pdSST-pdSIC) is forced with present-day SST and SIC. The difference between pdSST-pdSIC and piSST-piSIC is used to determine the response to SST warming and sea ice loss. The difference between piSST-pdSIC and piSST-piSIC is used to determine the response to only sea ice loss. Each model consists of 100–300 ensemble members each of one year with identical boundary conditions.

## Quantile regression and variability analysis

All data is first bilinearly interpolated to a common 2° latitude × 2° longitude grid before the analysis is performed. For the quantile regression, variance, and skewness calculations, we use the daily average near-surface temperatures without removing the seasonal cycle. We chose not to remove the seasonal cycle to focus on the absolute coldest days.

To calculate the trends as a function of percentile, we use quantile regression[16,44,45] with the daily averaged data. We calculate every second percentile (2nd, 4th, 6th, etc.). The results and conclusion are not sensitive to the specific choice of percentiles used for the analysis. To calculate variance and skewness trends, we first calculate the variance and skewness in each winter, which results in a 43-year time series for

each quantity at each grid point. To remove the variance trend from the quantile regression calculations, we use the approach described by McKinnon et al. [44], who used Legendre polynomials to map the trends as a function of percentile onto different moments. The trend linked with variance corresponds to the second Legendre polynomial, which consists of a linear change in percentile space. Zonal means of the trends were taken after computing trends at each grid point.

In the models, the quantile regression and variability calculations are performed on each ensemble member first. We then calculate the ensemble mean for each model before averaging over all models to calculate the multi-model mean. Thus, each model is weighted equally regardless of the number of ensemble members.

Most of the trends are divided by global, annual mean temperature trends so that trends are shown in the units of °C per °C of global warming. This was done because some of the models used here have a high climate sensitivity and show too much global warming relative to observations over the historical period[72]. Another reason is the different scenarios used for each model because of data availability. For the PAMIP experiments, the responses to both the pdSST-pdSIC and piSST-pdSIC experiments were divided by the global mean temperature response in the pdSST-pdSIC experiment.

Dividing the trend in each ensemble member by the global mean will alter the ensemble spread if the global mean trends are different from observations, making the comparisons between model spread and observations unfair. To analyse the model spread in Fig. 2, we applied the following procedure: We first remove the ensemble mean trend from each ensemble member. We then divide these residual trends (representing internal variability) by the observed global, annual mean temperature trend. This represents uncertainty due to internal variability (as estimated by model simulations) in the trend per degree of observed warming, allowing for a comparison with the trend per degree of warming in observations. Next, we divide the ensemble mean trend by the corresponding modelled global annual mean trend and add this to the residuals. Note this method assumes that global warming has little effect on internal variability over the 43-year time scales we examine. This method results in the proper scaling of the ensemble mean without artificially altering the ensemble spread.

### Uncertainty analysis

To estimate the uncertainty due to internal variability in the ERA5 trends, we apply a bootstrapping approach to resample the data. We use a block size of one season/year and resample the data 1000 times. The same resampling is done for each grid point, percentile, and moment. The resampled trends are added onto the ERA5 trends and the 2.5–97.5% range is displayed in Fig. 2. The stippling in Figs. 1a–e and 3a–c represent where the trends are not statistically significant after controlling for the false discovery rate[46] with $\alpha = 0.1$.

### Detection and attribution

We use a standard regression-based detection and attribution method[47,48]. We regress the observed trend pattern ($Y$) onto the model trend pattern or 'fingerprints' ($X_i$):

$$Y = \sum \beta_i X_i + \epsilon \tag{1}$$

where $\beta_i$ are the regression coefficients or 'scaling factors' and $\epsilon$ represent noise from internal variability. We perform a one-signal analysis using simple linear regression where the fingerprint is the multi-model mean trend pattern from the historical simulations. We also perform a three-signal analysis using multiple linear regression where the fingerprints are the set of three trend patterns (GHG, AER, NAT) from the multi-model single forcing ensemble large ensembles.

The uncertainty in the scaling factors is calculated by first removing the ensemble mean trends from each model, creating a total

of 300 trend patterns that arise from only internal variability. To account for biases that occur when subtracting the ensemble mean, the magnitude of each realization is scaled by $\sqrt{N/(N-1)}$, where $N$ is the number of realizations in the respective model. We then use the same regression model but replace the observations with each of these trend patterns arising from internal variability. This results in 300 scaling factors that are only due to internal variability. The 5–95th percentile range of these scaling factors is the uncertainty, and it is centred around the scaling factor. This method does not assume that the noise is normally distributed or that the noise is uncorrelated, but it does assume that the model accurately captures observed internal variability.

The magnitude of the scaling factors represents how much the model fingerprints must be scaled up or down to best match the observations. A scaling factor that is statistically above zero indicates that the modelled fingerprint is detectable in observations. A scaling factor consistent with one indicates that the magnitude of the model trend pattern is consistent with observations. A scaling factor greater than one indicates that the model fingerprint needs to be scaled up to best match the observations (i.e., the model underestimates the magnitude of the trend), and a scaling factor less than one means that the model fingerprint needs to be scaled down to best match the observations.

To be consistent with past studies, for the detection and attribution analysis we do not scale the trends by the global annual mean temperature. We note that if we divide the observed and model trends by the global mean trends and follow the procedure above to adjust the ensemble spread, the magnitude and uncertainty scaling factors change, but the signal-to-noise ratios remain constant. This means that whether a fingerprint is detectable or not is not sensitive to this choice.

## Data availability

All data used in this study is publicly available. The ERA5 data can be found at: https://cds.climate.copernicus.eu/cdsapp#!/dataset/reanalysis-era5-single-levels?tab=overview. JRA-55 data can be found at https://rda.ucar.edu/datasets/ds628.0/, NCEP-DOE reanalysis can be found at https://psl.noaa.gov/data/gridded/data.ncep.reanalysis2.html. Berkeley Earth gridded observational can be found at: https://berkeleyearth.org/data/. The ACCESS-ESM1-5, CanESM5, EC-Earth3, MIROC6, and MPI-ESM1-2-LR data can be found on the Earth System Grid Federation website: https://esgf-node.llnl.gov/projects/cmip6/. CESM2 large ensemble data can be found at https://www.cesm.ucar.edu/community-projects/lens2 and the single forcing large ensemble data from CESM2 can be found at https://www.cesm.ucar.edu/working-groups/climate/simulations/cesm1-single-forcing-le, GFDL-SPEAR-MED data can be found at: https://www.gfdl.noaa.gov/spear_large_ensembles/.

## Code availability

The quantile regression calculations were done using the Python package 'statsmodels' (https://www.statsmodels.org/stable/index.html). Plots with maps were done using the Python package 'cf-plot' (https://ajheaps.github.io/cf-plot/). Additional code to reproduce the figures of this study is available from the corresponding author.

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

## Acknowledgements
We thank the ECMWF for providing the ERA5 reanalysis and the modelling centres for performing the modelling experiments and making the data publicly available. We thank Michael Sigmond and Bill Merryfield for providing comments on the manuscript.

## Author contributions
R.B. conceived of the study, performed the analysis and wrote the paper. J.C.F. provided input on the analysis and commented on the manuscript.

## Competing interests
The authors declare no competing interests.
