## [Peer Review File · Nature Communications]

Amplified warming of North American cold extremes linked to human-induced changes in temperature variabilityREVIEWER COMMENTS

Reviewer #1 (Remarks to the Author):

Overall, this is a well-written and interesting paper that I think can be accepted for publication after some minor revision. It does not present entirely new results but reinforces previous studies, and confirms the role of human induced greenhouse gases on previously projected temperature variability changes (which are here analyzed for historical data too).

A few minor comments are summarized below:

- Line 30: kurtosis is also very important for the higher temperature variability changes and should be noted (e.g., <https://doi.org/10.1175/JCLI-D-21-0310.1>)

- Line 36-38: You can add why this is happening (why are the cold anomalies weakening more)

-Line 50-51: again, kurtosis is important too.

-Lines 78-79: add which level

-Line 85: daily mean?

- how do the results shown in Fig.1 compare with the actual skewness changes? Maybe add the skewness map to the SI?

-Line 130: “which is confirmed be calculated...”

- line 158: as function latitude and longitude  as a function latitude and longitude

- Line 186: Note that the arguments discussing the temperature gradient typically refer to the 850hPa, i.e., above the boundary layer.

-Figure 1: add which level

Reviewer #2 (Remarks to the Author):

Review of "Amplified warming of North American cold extremes linked to human-induced changes in temperature variability"

Summary

It is already well-established that wintertime daily temperature variance is decreasing over North America due to polar amplification. The most important result of this paper is the demonstration that similar patterns are simulated by models, and that they can be detected in various ways in a formal D&A analysis (although see my major comment on this point). The contribution of sea ice changes are also shown through analysis of PAMIP experiments. The analyses are all reasonable, and the results make sense, and are consistent with prior work. Given that the paper is primarily a formal confirmation of existing work, I would generally think it would be a better fit for e.g. Journal of Climate.

Major comments

- I think a more complete accounting for uncertainty and internal variability is necessary. (1) There is no uncertainty provided for the ERA5 trends in Figure 1 or Supp Figure 2. Uncertainty can be estimated by bootstrapping, and is particularly important for Supp Figure 2, where the spread across models indicates that the uncertainty in these trends due to sampling of internal variability is large. (I would interpret a bootstrap-based estimate of uncertainty as the uncertainty due to resampling of atmospherically-driven variability without substantial interannual memory.) (2) As shown, Supp Fig 2 more or less suggests that colder, average, and warm temperatures are all warming, within uncertainty, at the same rate. I suspect that this is not the case, but rather that a member with a relatively low mean warming also has lower warming in the extremes. It would thus be more helpful to show the spread of 98-50 and 2-50. (See my minor comment below about using less extreme quantiles as well.) This updated figure should be combined with Figure 1 in the main text to provide the reader with information about internal variability.

- Figure 1 (especially panels d, e, i, j) bring to mind the signal-to-noise paradox: are the observations closer to the EM than any individual simulation is?

- The discussion of Figure 4 does not reflect the figure very well. In my understanding of D&A, a beta value around 1 means that the observations exhibit the same signal with a similar magnitude as the models, whereas a $\beta > 1$ indicates that the model signal is too weak. It seems to me that "The variance trends are highly detectable" should really be "the variance trends are detectable, but likely too weak in the models". For skewness, the lower bound barely overlaps with zero but its mean value is close to one, both of which are important even if it is not "officially" detectable at a specified level. It will always be harder to detect changes in higher order moments, so providing some indication of how much of the distribution is positive would be helpful.

- In the D&A analysis, what method are the authors using to model the noise? It should not be

considered uncorrelated and normally distributed as would be done for standard OLS regression.

Minor comments

- Line 43: "would suggest" -> "means" (these two things are equivalent, unless there are very large changes in the higher-order moments)
- Line 48: Rhines et al 2017, "Seasonally Resolved Distributional Trends of North American Temperatures Show Contraction of Winter Variability" shows exactly how changes in the tails compare to changes in the mean, although the analysis only goes through 2014.
- Line 81: It's unclear to me what is meant by dividing my global, annual mean temperature. Does this mean that the authors are fitting linear trends, and then dividing by the change in temperature over the period? If so, a better way to do the analysis is to simply regress on GMT, which is an increasingly common approach to try to identify forced trends. GMT should be low-pass filtered to reduce the influence of internal variability.
- Figure 1 is so small (and pixelated in my copy) that it is hard to see. I suggest making it higher resolution, perhaps having fewer columns, and removing the contours, which are very dense in the upper row.
- Line 85: Quantile regression trends are increasingly variable at high/low percentiles. I suggest using the 5th/95th to reduce the noise in the estimates.
- For consistency with the quantile framework, I suggest show trends in the median throughout, rather than mean.
- Lines 126 and 142: How are the authors removing the variance trend, and mapping onto trends in the various moments? As noted in Rhines et al (2017), "The mapping of the quantile trends shown in Fig. 2 onto moments such as variance and skewness depends on the base distribution". They could instead use the Legendre polynomials as suggested in McKinnon et al (2016), The changing shape of Northern Hemisphere summer temperature distributions. Updated: I now see that the authors assume normality when they remove the contribution of the mean and variance. This seems somewhat strange given the focus on nonnormality and a quantile framework.

Reviewer #3 (Remarks to the Author):

Review of "Amplified warming of North American cold extremes linked to human-induced changes in temperature variability" by Blackport and Fyfe submitted to Nature Communications

The manuscript presents a comprehensive analysis of how cold extremes are changing in North America through quantile regressions. It effectively delves into the connection with Arctic amplification, providing a detailed analysis and attributing trends to specific factors. The clarity and quality of the writing make it valuable for both the scientific community and decision-makers, supporting my recommendation for its publication. However, I would like to share a few suggestions and address some uncertainties, which are outlined below:

- 1) Figures 1 and 2: To enhance clarity in the presentation of observed trends in both mean and percentiles, consider indicating or visually representing only the regions where these trends reach statistical significance.
- 2) Lines 94 and 103: The phrase "warming slower than the mean" might suggest that the 98th percentile is experiencing a slower rate of warming compared to the mean. However, in numerous regions of Canada, this percentile is actually decreasing, implying a cooling trend. To enhance clarity, the authors may consider rephrasing this expression to accurately convey the observed cooling phenomenon in these regions. It might be worth mentioning that the mean and the 98th percentile show trends of opposite signs.
- 3) Section "The changing shape of winter temperature distributions": In the Supplementary Material, it might be beneficial to include a figure illustrating the empirical distribution functions of winter temperatures during the initial 20 years and the last 20 years. Creating separate visuals for the region north of 45°N and south of 45°N would provide a more visual representation of the changes in distribution over time, enhancing the readers' understanding of the evolving patterns.
- 4) Figure 3: To facilitate a more straightforward comparison, it is recommended to use the same scale on the x-axis. This consistency will allow readers to easily assess and contrast the data across different elements of the figure.
- 5) Lines 165-166 and 225-227: The statements in these sentences are somewhat confusing. Figure 4 doesn't explicitly convey whether the models adequately represent the observed trends. The values exceeding 1 imply an underestimation of the trend in variance by the models. Additionally, Supplementary Figure 3 suggests that the multi-model mean underestimates variance trends. Clarification or revision of these sentences may help in conveying the nuanced relationship between model outputs and observed trends in variance.
- 6) Would it not be more interesting to move Figure 4 to the Supplementary Material and Supplementary Figure 5 to the main text and analyse the latter in detail?
- 7) Caption of Figure 5 is a bit confusing with so many "response", please try to rewrite it a bit clearer.
- 8) Lines 269-270: It would be beneficial to add that there are a total of 300 realisations. This additional information will provide context and clarity when this specific number is referenced later in the manuscript.
- 9) Lines 324-326: The current explanation of how this procedure effectively eliminates the variance trend may lack clarity. Please consider providing a more detailed explanation to enhance understanding of how this particular procedure serves to remove the variance trend.

We thank all three reviewers for taking the time to review our paper and provide constructive comments which have improved the manuscript. The main changes we have made are: (1) A more detailed analysis and discussion of uncertainty, and (2) a number of improvements to the presentation of the figures. The reviewers' comments are copied below in black, and our responses are in blue.

Reviewer #1 (Remarks to the Author):

Overall, this is a well-written and interesting paper that I think can be accepted for publication after some minor revision. It does not present entirely new results but reinforces previous studies, and confirms the role of human induced greenhouse gases on previously projected temperature variability changes (which are here analyzed for historical data too).

We thank the reviewer for their positive comments. We are pleased to see that the reviewer finds the paper interesting and recommended publication. As to the reviewer's second point, we have taken the opportunity in revision to make clearer the originality of our results. As we explain in more detail to reviewer #2, the direct model and observation comparisons and the focus on the higher moments of the temperature distributions in observed trends are (to our knowledge) entirely new.

A few minor comments are summarized below:

- Line 30: kurtosis is also very important for the higher temperature variability changes and should be noted (e.g., <https://doi.org/10.1175/JCLI-D-21-0310.1>)

Added.

- Line 36-38: You can add why this is happening (why are the cold anomalies weakening more)

We have added more detail about why the cold anomalies are weakening more at Lines 37-39.

-Line 50-51: again, kurtosis is important too.

We agree, so we have added kurtosis as well. The study linked above is the only one we could find that examines projections of the kurtosis of the winter temperature. They find a decrease in kurtosis, which would be expected to be linked with an even stronger warming of extreme cold temperature (but they also find that the changes in kurtosis are strongly linked to the skewness changes).

-Lines 78-79: add which level

We have added that we examined near surface temperature.

-Line 85: daily mean?

Changed to daily mean.

- how do the results shown in Fig.1 compare with the actual skewness changes? Maybe add the skewness map to the SI?

The maps of the skewness change are shown in Supplementary Fig. 4. These are consistent with the maps in Fig 1, with increased skewness south of 40°N where we also see stronger warming of the coldest days without weaker warming of the warmest days. This is discussed later when discussing Figure 3 and 4, so we now mention that we will return to this in the next section.

-Line 130: "which is confirmed be calculated..."

Fixed.

- line 158: as function latitude and longitude  as a function latitude and longitude

Fixed.

- Line 186: Note that the arguments discussing the temperature gradient typically refer to the 850hPa, i.e., above the boundary layer.

We have added that this was referring to temperatures above the boundary layer.

-Figure 1: add which level

Added that it is near-surface temperature.

Reviewer #2 (Remarks to the Author):

Review of "Amplified warming of North American cold extremes linked to human-induced changes in temperature variability"

Summary

It is already well-established that wintertime daily temperature variance is decreasing over North America due to polar amplification. The most important result of this paper is the demonstration that similar patterns are simulated by models, and that they can be detected in various ways in a formal D&A analysis (although see my major comment on this point). The contribution of sea ice changes are also shown through analysis of PAMIP experiments. The analyses are all reasonable, and the results make sense, and are consistent with prior work. Given that the paper is primarily a formal confirmation of existing work, I would generally think it would be a better fit for e.g. Journal of Climate.

We thank the reviewer for their constructive comments that have significantly improved the paper.

As to the reviewer's final point, we would like stress our view that the results we present are much more than a formal confirmation of existing work. First, the like-for-like comparisons between observed and model trends over recent decades is novel. This is particularly important because there have been a

number of high-profile papers and reviews that have questioned whether models accurately capture trends in extreme cold over the midlatitudes (e.g. Cohen et al. 2020, 2022), which has important implications for future projections. Importantly however, these papers have not directly compared models and observations. Instead, they are based off comparing the conclusions of model-based studies and observation-based studies. The strong agreement between models and observations when they are compared like-for-like, indicates that the any disagreements (at least over North America) are likely a results of misinterpretation observations rather than model errors.

Another novel result is the link between extreme cold and changes in the higher moments of distributions in observed trends. Rhines et al. (2017) examined trends in winter extremes using quantile regression in observations, but there was essentially no discussion of the higher moments beyond changes in variance. Our results show that these higher moments appear to be also playing a role in the observed trends over some regions, consistent with model projections.

We have made some additions to the introduction and conclusions (Lines 54-56,76-77,277-279), to better highlight the novelty and importance of our work.

Major comments

- I think a more complete accounting for uncertainty and internal variability is necessary. (1) There is no uncertainty provided for the ERA5 trends in Figure 1 or Supp Figure 2. Uncertainty can be estimated by bootstrapping, and is particularly important for Supp Figure 2, where the spread across models indicates that the uncertainty in these trends due to sampling of internal variability is large. (I would interpret a bootstrap-based estimate of uncertainty as the uncertainty due to resampling of atmospherically-driven variability without substantial interannual memory.) (2) As shown, Supp Fig 2 more or less suggests that colder, average, and warm temperatures are all warming, within uncertainty, at the same rate. I suspect that this is not the case, but rather that a member with a relatively low mean warming also has lower warming in the extremes. It would thus be more helpful to show the spread of 98-50 and 2-50. (See my minor comment below about using less extreme quantiles as well.) This updated figure should be combined with Figure 1 in the main text to provide the reader with information about internal variability.

Thanks for the excellent suggestion. We have done the uncertainty analysis of the ERA5 trends using a bootstrap approach, as suggested by the reviewer. We have resampled the data 1000 times with a block size of one season/year. The same resampling is done for each grid point, percentile, and moment.

The bootstrap samples are used to estimate the uncertainty of the spatial averages, plotted as the 2.5-97.5 percentile range in former Supplementary Fig 2, which we have moved to the main paper (now Fig 2). This shows that the trends in the 2nd percentile trends averaged over the United States and Southern Canada are statistically significant. To be more consistent with the ERA5 uncertainty range, we now only show the 2.5-97.5% range in the boxplots for the models (we previously showed the min to max range).

We have also added another panel in Fig 2 to show the 2nd – mean and 98th – mean trends, their uncertainties, and their ensemble spreads. These plots indicate that the difference between 2nd and mean averaged over the US and Southern Canada is marginally statistically significant. The difference between 98th and the mean is not statistically significant for this region, but it is over other regions(e.g., over Northern Canada; not shown).

We have also added stippling to the observed Figure 1a-e trends to show the grid points that are not statistically significant as determined by the bootstrap approach.

We have added some discussion of the uncertainties in observations and model spread shown in Fig 2 at Lines 119-131.

- Figure 1 (especially panels d, e, i, j) bring to mind the signal-to-noise paradox: are the observations closer to the EM than any individual simulation is?

This is an interesting question. It is not entirely clear what is meant by 'closer', but we interpret this as similarity in the pattern, which can be calculated using a pattern correlation. We have calculated the pattern correlation between the ensemble mean trend pattern and both observed trend pattern and the patterns in each individual model realization. The results for the differences between extreme and mean trends are shown below in Figure R1. We find that that the pattern correlation with observations is slightly higher than the average correlation from the model ensemble, but they are well within the ensemble distribution. Thus, there does not appear to be evidence for a signal-to-noise paradox in these trends.

Figure R1: The pattern correlation of the ERA5 trend with multi-model mean for the 2nd percentile – mean trends (blue dot) and the 98th percentile – mean (red dot). The box and whisker plots represent the ensemble spread of the pattern correlations in individual ensemble members with the multi-model mean. The box represents the inner quartile range, the line represents the median, and the whiskers represent the minimum and maximum pattern correlation across all realizations.

- The discussion of Figure 4 does not reflect the figure very well. In my understanding of D&A, a beta value around 1 means that the observations exhibit the same signal with a similar magnitude as the models, whereas a beta > 1 indicates that the model signal is too weak. It seems to me that "The variance trends are highly detectable" should really be "the variance trends are detectable, but likely too weak in the models". For skewness, the lower bound barely overlaps with zero but its mean value is close to one, both of which are important even if it is not "officially" detectable at a specified level. It will

always be harder to detect changes in higher order moments, so providing some indication of how much of the distribution is positive would be helpful.

We agree that our description of this figure could have been better. We agree that the scaling factor greater than one could indicate the models are underestimating the variance trends. However, we should note that uncertainty overlaps with one (8% of the distribution is below one), suggesting that the stronger observed trends could be a result of an unusually strong trend due to internal variability, and not model error. Also, there could potentially be some selection bias as we chose to focus on North America, in part, because of the known large decreasing trends in variance that had been previously identified. We have added a more detailed description that models may underestimate the variance trends (Lines 192-196).

For skewness, we agree that it would be important to point out that the scaling factor is close to one and that it is close to being 'officially' detectable. About 7% percent of the distribution is below one, which corresponds to a p-value of 0.07. We have added more details about this at Lines 196-199.

- In the D&A analysis, what method are the authors using to model the noise? It should not be considered uncorrelated and normally distributed as would be done for standard OLS regression.

As we describe in the methods, the noise is calculated from internal variability in the models, as has been done in many previous D&A studies (e.g. Swart et al. 2018). Specifically, we remove the ensemble mean trends in each model so that we have 300 trends pattern (from the 300 total realizations) that are entirely internal variability/noise. We then repeat the regression, but we replace the observations with each of these 300 independent trend patterns that represent the noise. The result of this is a range of 300 scaling factors that are entirely due to internal variability/noise. The uncertainty in the scaling factors plotted in Fig. 5 is the 5-95th % range.

This method does not assume that noise is normally distributed or that it is uncorrelated, but it does assume that the models accurately capture the noise (as is often the case in such fingerprint D&A analysis).

We have added more details about this in the main text (Lines 182-185) and the methods (Lines 419-421).

Minor comments

- Line 43: "would suggest" -> "means" (these two things are equivalent, unless there are very large changes in the higher-order moments)

We agree that these two should be essentially equivalent, but the reviewer's caveat that "unless there are very large changes in the higher-order moments" is important. As we describe in the following paragraph, there have been several recent high-profile papers(e.g Cohen et al. 2021) suggesting that extreme cold over North America is becoming more frequent and intense because of changes in atmospheric circulation. This appears to be at odds with the well-established decrease in temperature variance that has been observed over much of North America. One of the goals of this study is to assess whether the extreme cold trends are consistent with expectations from trends in mean and variance, or

whether higher moments also play an important role. We find that they do, but they are actually linked with even stronger reductions in extreme cold.

We strengthened this by changed to “would suggest” to “would strongly suggest”.

- Line 48: Rhines et al 2017, "Seasonally Resolved Distributional Trends of North American Temperatures Show Contraction of Winter Variability" shows exactly how changes in the tails compare to changes in the mean, although the analysis only goes through 2014.

We have changed this to more accurately reflect the literature, including a more explicit reference to Rhines et al. 2017.

- Line 81: It's unclear to me what is meant by dividing my global, annual mean temperature. Does this mean that the authors are fitting linear trends, and then dividing by the change in temperature over the period? If so, a better way to do the analysis is to simply regress on GMT, which is an increasingly common approach to try to identify forced trends. GMT should be low-pass filtered to reduce the influence of internal variability.

Yes, we have fitted the linear trends, then divided these trends by the global mean temperature trends. We have clarified this at Lines 88-89.

The approach proposed by the reviewer is a good suggestion, but for our purpose, it will likely make little difference in the results. Our approach is essentially equivalent, but with a very low pass filter, such that the GMT is just the linear trend. We note that over the analysis period (1980-2022), the global mean temperature increase is approximately linear. There could be a difference if longer periods are analyzed, when the GMT changes show more non-linear behaviour.

The approach suggested by the reviewer also introduces additional issues with quantifying ensemble spread. If models show a magnitude of global warming substantially different to that of the observed trends (which is the case for a few of the models), either dividing my global warming trends or regressing on GMT, will result in the ensemble spread being altered (because the magnitude of internal variability will also be scaled by GMT trends). With our approach, we can correct for this (as detailed in Lines 378-389 in the methods), but it is not clear how this could be done when regressing on GMT. For these reasons we decided to keep the current methods used.

- Figure 1 is so small (and pixelated in my copy) that is is hard to see. I suggest making it higher resolution, perhaps having fewer columns, and removing the contours, which are very dense in the upper row.

We agree that the panels in Fig 1 were a bit too small and hard to see. We have rearranged the Figure into 3 columns and bigger panels. This rearrangement also is bit more logical with the mean on left, the 2nd and 98th percentile in the middle and the differences on the right. We have also removed the contour lines as suggested.

- Line 85: Quantile regression trends are increasingly variable at high/low percentiles. I suggest using the 5th/95th to reduce the noise in the estimates.

Thanks for the suggestion. Supplementary Fig. 3 repeats the new Fig 2, but with the of less extreme percentiles, specifically the 6th and 94th (we didn't have the 5 and 95th already calculated in the models). While these are less noisy, the trends are weaker, particularly for the extremely cold percentiles. This is likely linked to the changes in skewness. The weaker, but less noisy trends result in signal-to-noise ratios that are actually similar to the temperature trends at the more extreme percentiles. As the more extreme temperatures have greater impacts and better highlight the higher moments, we have kept the analysis of the 2nd and 98th percentile in the main paper, but included some analysis of the 6th and 94th in the supplementary figures and discuss this briefly at Lines 129-131.

- For consistency with the quantile framework, I suggest show trends in the median throughout, rather than mean.

We chose to focus on the mean instead of the median/50th percentile trends because it is much more common to analyze the mean trends than it is the median/50th percentile. Because it is more common, we think it is more relevant to compare the extremes to how the mean is changing. Also, the use of mean is more consistent with the analysis of the other moments.

We have repeated Figure 1 but using the 50th percentile instead of the mean (Supplementary Fig. 2) The differences between the mean and median/50th percentile trends are small, so it makes little difference in our results and conclusions.

- Lines 126 and 142: How are the authors removing the variance trend, and mapping onto trends in the various moments? As noted in Rhines et al (2017), "The mapping of the quantile trends shown in Fig. 2 onto moments such as variance and skewness depends on the base distribution". They could instead use the Legendre polynomials as suggested in McKinnon et al (2016), The changing shape of Northern Hemisphere summer temperature distributions. Updated: I now see that the authors assume normality when they remove the contribution of the mean and variance. This seems somewhat strange given the focus on nonnormality and a quantile framework.

We thank the reviewer for pointing to an alternative way to remove the variance trends using Legendre polynomials. We now use this method to calculate and remove the trends/changes that are linked to changes in variance. Note that the results are very similar to the previous analysis where we assumed normality, and this does not change any of the conclusions.

Reviewer #3 (Remarks to the Author):

Review of "Amplified warming of North American cold extremes linked to human-induced changes in temperature variability" by Blackport and Fyfe submitted to Nature Communications

The manuscript presents a comprehensive analysis of how cold extremes are changing in North America through quantile regressions. It effectively delves into the connection with Arctic amplification, providing a detailed analysis and attributing trends to specific factors. The clarity and quality of the writing make it valuable for both the scientific community and decision-makers, supporting my recommendation for its publication. However, I would like to share a few suggestions and address some uncertainties, which are outlined below:

We thank the reviewer for the positive and constructive comments which have helped improve the manuscript.

1) Figures 1 and 2: To enhance clarity in the presentation of observed trends in both mean and percentiles, consider indicating or visually representing only the regions where these trends reach statistical significance.

We have added stippling to the observed trends in Fig. 1 and what is now Fig. 3 that indicate where the trends are not statistically significant. Statistical significance is calculated using the bootstrap approach suggested by reviewer #2 and described at Lines 391-398. We have also rearranged Fig. 1 and enlarged the panels to improve the appearance. We have also changed the contour levels in Fig. 3, so that there are fewer levels.

2) Lines 94 and 103: The phrase "warming slower than the mean" might suggest that the 98th percentile is experiencing a slower rate of warming compared to the mean. However, in numerous regions of Canada, this percentile is actually decreasing, implying a cooling trend. To enhance clarity, the authors may consider rephrasing this expression to accurately convey the observed cooling phenomenon in these regions. It might be worth mentioning that the mean and the 98th percentile show trends of opposite signs.

We have now highlighted that the ERA5 98th percentile trends are negative and opposite to the mean, but not statistically significant (Lines 98). The "warming slower than the mean" was in reference to the model trends which is accurate. We have now clarified this (Lines 114-115).

3) Section "The changing shape of winter temperature distributions": In the Supplementary Material, it might be beneficial to include a figure illustrating the empirical distribution functions of winter temperatures during the initial 20 years and the last 20 years. Creating separate visuals for the region north of 45°N and south of 45°N would provide a more visual representation of the changes in distribution over time, enhancing the readers' understanding of the evolving patterns.

Thanks for the suggestion. We agree it would be helpful to show the distributions for visualization purposes. However, instead of showing the temperature distributions for the regions, we chose to plot the distributions for representative grid points for each of the two regions. This is because within each region, the underlying distribution can be different, even if the changes in extremes and moments are similar. When the temperature anomalies across grid points are combined, the changes in the distribution can get smoothed out and hard to see. This is especially the case for the lower latitudes where the higher moments are the most important.

We have included the temperature distributions for the two time periods for representative grid points (55°N,99°W and 37°N and 91°W) in Supplementary Fig. 6 and refer to it at Lines 169-171.

4) Figure 3: To facilitate a more straightforward comparison, it is recommended to use the same scale on the x-axis. This consistency will allow readers to easily assess and contrast the data across different elements of the figure.

We now use the same x-axis on both panels of what is now Figure 4, as suggested.

5) Lines 165-166 and 225-227: The statements in these sentences are somewhat confusing. Figure 4 doesn't explicitly convey whether the models adequately represent the observed trends. The values exceeding 1 imply an underestimation of the trend in variance by the models. Additionally, Supplementary Figure 3 suggests that the multi-model mean underestimates variance trends. Clarification or revision of these sentences may help in conveying the nuanced relationship between model outputs and observed trends in variance.

We agree some of this was a bit confusing originally. We have added more detail about the variance trends at Lines 193-196. See also the response to reviewer #2 who had a similar comment.

6) Would it not be more interesting to move Figure 4 to the Supplementary Material and Supplementary Figure 5 to the main text and analyse the latter in detail?

After some consideration we decided to leave Fig. 5 (formally Fig. 4) in the main paper. This is because Supplementary Fig. 7 (formally 5) is a lot less "clean" and more different to the rest of the paper than the analysis presented in Figure 5. Only 3 of the 7 models have the large ensemble single forcing runs, and one of those (CESM2) does not have a 'natural only' run. Instead, CESM2 has an 'everything else' run that includes additional forcing (e.g. land cover changes), that we treat as a 'natural only run.' Also, some of these have fewer ensemble members (as few as 10) which results in greater uncertainty in the forced response. For these reasons, we do not want to put too much focus on these results. However, they are consistent with the responses primarily being driven by greenhouse gas emissions, so we think it is worth briefly mentioning.

7) Caption of Figure 5 is a bit confusing with so many "response", please try to rewrite it a bit clearer.

We agree this was confusing and hard to read. We have edited the caption for clarity.

8) Lines 269-270: It would be beneficial to add that there are a total of 300 realisations. This additional information will provide context and clarity when this specific number is referenced later in the manuscript.

Added.

9) Lines 324-326: The current explanation of how this procedure effectively eliminates the variance trend may lack clarity. Please consider providing a more detailed explanation to enhance understanding of how this particular procedure serves to remove the variance trend.

Reviewer #2 pointed to a better method to remove the trends that uses Legendre polynomials. We have added this to the methods section and cited the appropriate paper that describes the methods and justification in detail (Lines 360-363).

References

- Cohen, J., and Coauthors, 2020: Divergent consensuses on Arctic amplification influence on midlatitude severe winter weather. *Nat. Clim. Chang.*, **10**, 20–29, <https://doi.org/10.1038/s41558-019-0662-y>.
- Cohen, J., L. Agel, M. Barlow, C. I. Garfinkel, and I. White, 2021: Linking Arctic variability and change with extreme winter weather in the United States. *Science*, **373**, 1116–1121, <https://doi.org/10.1126/science.abi9167>.
- Cohen, J., L. Agel, M. Barlow, C. I. Garfinkel, and I. White, 2022: Arctic change reduces risk of cold extremes—Response. *Science*, **375**, 729–730, <https://doi.org/10.1126/science.abn8954>.
- Rhines, A., K. A. McKinnon, M. P. Tingley, and P. Huybers, 2017: Seasonally Resolved Distributional Trends of North American Temperatures Show Contraction of Winter Variability. *Journal of Climate*, **30**, 1139–1157, <https://doi.org/10.1175/JCLI-D-16-0363.1>.
- Swart, N. C., S. T. Gille, J. C. Fyfe, and N. P. Gillett, 2018: Recent Southern Ocean warming and freshening driven by greenhouse gas emissions and ozone depletion. *Nature Geosci*, **11**, 836–841, <https://doi.org/10.1038/s41561-018-0226-1>.

REVIEWER COMMENTS

Reviewer #2 (Remarks to the Author):

Second review of "Amplified warming of North American cold extremes linked to human-induced changes in temperature variability"

I thank the authors for their responses to my prior review. I have a few minor comments, and a broad comment on the framing for the authors and Editor to consider.

While I now more clearly understand the authors' motivation as counteracting the results presented in the various Cohen papers, I would argue those papers are outliers, and the scientific consensus is already behind the idea that polar amplification leads to reduced NH midlatitude temperature variance as first outlined in Screen (2014) Nature Climate Change (<https://www.nature.com/articles/nclimate2268>) and discussed in other more recent papers cited by the authors. Along these lines, while Judah Cohen and colleagues (Jennifer Francis in particular) may agree with the statement "How global warming is impacting winter cold extremes is uncertain", I don't think almost any other climate scientist would, and I suggest moderating it in the abstract given the reasonably large number of papers already pointing out how cold winter days are warming quickly. It seems disingenuous to view the Cohen and Francis view as on balance with numerous papers across multiple groups in terms of how cold days are changing.

Minor

- In their reply, the authors suggest novelty in that they consider the role of different moments in causing changes in extremes. While this may be technically true, it's not clear to me what additional insight this provides. While high/low quantiles provide a direct quantification of how extremes are changing, one cannot immediately map from moment changes to changes in extremes unless all moments are moving in the same direction. What is the additional scientific insight from understanding how the moments change beyond what you would get from looking at quantiles?
- In general, the authors should account for multiple hypothesis testing (across gridboxes, across longitude/percentile space) through e.g. controlling for a false discovery rate.
- line 90: The region of weak warming appears to be at least half of the domain
- line 150: To me, the more notable trend south of 45N is the switch where the moderately cold days are warming less than mean, which also shows up in Figure 3c after removing the variance trends. This is less relevant for the extremes, but would be relevant here because the authors are discussing the mapping from quantile trends to moments (although see first minor comment above).
- line 173: I suspect that the greater contribution of the mean for the multi-model mean in Figure 4 is because internal variability is being averaged out. Do the authors want to mention this explicitly? Otherwise a reader may interpret this as reflecting a model-observations difference, which I don't think is the intent, or is correct.

- line 225: it should be noted (and perhaps commented on if the authors know why) that the greater warming of the cold extremes than the mean is shifted further south in both PAMIP experiments.

- line 280: where is this shown?

- line 351: please state the motivation to not remove the seasonal cycle. Is it to focus on the absolutely coldest, rather than relatively coldest, days?

We thank the reviewer for their constructive comments. We have copied the reviewer's comments below in black and our responses are in blue.

Reviewer #2 (Remarks to the Author):

Second review of "Amplified warming of North American cold extremes linked to human-induced changes in temperature variability"

I thank the authors for their responses to my prior review. I have a few minor comments, and a broad comment on the framing for the authors and Editor to consider.

While I now more clearly understand the authors' motivation as counteracting the results presented in the various Cohen papers, I would argue those papers are outliers, and the scientific consensus is already behind the idea that polar amplification leads to reduced NH midlatitude temperature variance as first outlined in Screen (2014) Nature Climate Change (<https://www.nature.com/articles/nclimate2268>) and discussed in other more recent papers cited by the authors. Along these lines, while Judah Cohen and colleagues (Jennifer Francis in particular) may agree with the statement "How global warming is impacting winter cold extremes is uncertain", I don't think almost any other climate scientist would, and I suggest moderating it in the abstract given the reasonably large number of papers already pointing out how cold winter days are warming quickly. It seems disingenuous to view the Cohen and Francis view as on balance with numerous papers across multiple groups in terms of how cold days are changing.

We agree with the reviewer that the evidence for reduced temperature variability is already strong and that the evidence Cohen et al. present regarding cold extremes is not convincing. However, we disagree that the Cohen et al. studies are outliers, and we disagree that there is already a scientific consensus that strongly supports the strong warming of extreme cold days.

First of all, Cohen et al. continue to publish papers in high-impact journals arguing that cold extremes over North America (and the midlatitudes more generally) are becoming more severe/frequent, or at least not reducing in severity/frequency as fast as models predict (Cohen et al. 2018, 2020, 2021, 2022, 2023). This includes a new study that was published shortly after our paper was submitted claiming that there is no detectable trend in midlatitude cold extremes (Cohen et al. 2023). There was also a high-profile review paper on the influence of Arctic warming (and implicitly, global warming) on extreme cold winter weather that highlights uncertainties and disagreement among studies, particularly among observational vs model studies (Cohen et al. 2020). This review paper has 30+ authors and made it through peer review, so it is certainly not only Cohen and Francis that would agree that there is uncertainty in how extreme cold might change in warming climate. These studies continue to be highly influential and receive a very high number of citations (the overwhelming majority of which are not critical), both within the physical climate science and impacts literature (in addition to the disproportionate media attention).

Second, the lack of consensus is also highlighted in recent assessment reports, including the 5th National Climate Assessment report (NCA5). The work of Cohen et al. is very influential in the conclusions about the influence of global warming on cold extremes over the United States. The report even states cold extremes may becoming more likely, but that there is a lot of uncertainty. This is illustrated by this quote

from Chapter 5, Key Message 5.1, Uncertainties and Research Gaps (with the numbered references replaced):

“Whereas emerging research suggests that the frequency of cold-weather events and heavy snowfall may be increasing because of warming Arctic temperature (Cohen et al. 2018), there is some disagreement in the research community (Blackport et al. 2019; Cohen et al. 2020) regarding this projection...”

See also the discussion in Chapter 2 on ‘Arctic Changes Affect Weather in the Midlatitudes’ and Chapter 3 on ‘Extreme Heat and Cold’ in the NCA5 which also emphasize the uncertainty of how extreme cold is changing in response to increase global and Arctic temperatures.

The above would strongly suggest that the impact of global warming on cold extremes is still considered uncertain in the wider community. The work of Cohen et al. continues to be highly influential, so we believe it is important to address the claims that are often made in these studies regarding how extreme cold temperatures are changing and the apparent discrepancies between observations and models. Our study was in part motivated by the response of Cohen et al. (2022) which explicitly makes the claim that climate models are overestimating the warming of cold extremes (albeit without presenting evidence). Our study also rules out the possibility that large changes in higher moments of the temperature distribution could potentially result in increase in cold extremes despite a decrease in variance, which could have potentially reconciled the opposing views.

In summary, while this reviewer may think the Cohen et al. papers are not convincing and are outliers, this does not appear to be the consensus within the climate science community. We therefore think that our motivation is justified and think it is important to address the claims raised by Cohen et al. studies (and others that come to similar conclusions).

We have made a few minor changes to the abstract and introduction to highlight that the uncertainty continues to exist (Lines 12 and 68-70). We have also added references to the most recent Cohen et al. (2023) study and the NCA5.

Minor

- In their reply, the authors suggest novelty in that they consider the role of different moments in causing changes in extremes. While this may be technically true, it's not clear to me what additional insight this provides. While high/low quantiles provide a direct quantification of how extremes are changing, one cannot immediately map from moment changes to changes in extremes unless all moments are moving in the same direction. What is the additional scientific insight from understanding how the moments change beyond what you would get from looking at quantiles?

We think it is important to connect the changes in extremes and quantiles because there are a lot of studies that only examine the changes in mean and/or variance and implicitly assume that changes in extremes reflect the changes in these moments. By making the link between the moments and quantiles we can test this and are able to conclude that the mean and variance alone are not enough to infer changes in cold extremes in some regions. We have added a sentence to highlight this point in the conclusions (Line 265-266).

- In general, the authors should account for multiple hypothesis testing (across gridboxes, across longitude/percentile space) through e.g. controlling for a false discovery rate.

We have now controlled for the false discovery rate when calculating statistical significance in grid boxes (Fig 1) and latitude/percentile space (Fig 3). After accounting for this, there are few statistically significant trends at individual points, which we now mention (Lines 109-111). We note, however, that our study never depended on the statistical significance at individual grid box, which is why we barely mentioned it in previous versions of the manuscript. More importantly, the regional averages and the pattern detection and attribution results are statistically significant, so lack of statistical significance at individual grid boxes does not impact the conclusions of the study.

- line 90: The region of weak warming appears to be at least half of the domain

We agree, so we have removed that it is a “small” region.

- line 150: To me, the more notable trend south of 45N is the switch where the moderately cold days are warming less than mean, which also shows up in Figure 3c after removing the variance trends. This is less relevant for the extremes, but would be relevant here because the authors are discussing the mapping from quantile trends to moments (although see first minor comment above).

We now point out the weak cooling of the moderate days that are linked with the higher moments (Lines 157). However, as our focus is on the extremes, we have not discussed this in detail.

- line 173: I suspect that the greater contribution of the mean for the multi-model mean in Figure 4 is because internal variability is being averaged out. Do the authors want to mention this explicitly? Otherwise a reader may interpret this as reflecting a model-observations difference, which I don't think is the intent, or is correct.

We agree and it was not our intention to suggest that differences between model and observations are due to model error. We now explicitly state that these differences could be a result of internal variability and are not necessarily due to model error (Lines 179-182).

- line 225: it should be noted (and perhaps commented on if the authors know why) that the greater warming of the cold extremes than the mean is shifted further south in both PAMIP experiments.

We have now noted this (Line 235-236). We don't have an explanation for this, but we speculate that the fixed SSTs/sea ice in the PAMIP experiments may play a role. The fixed SST/sea ice may make it harder for extreme temperatures to occur over and near the ocean/sea ice. At high latitudes, where the largest differences are seen at high latitudes, most grid boxes are close to the ocean sea ice.

- line 280: where is this shown?

This is in reference to the fact cold extremes are generally warming faster than the mean temperature, and in some regions even have the opposite sign trend (e.g. Fig 1). This is an important point because some studies have used the lack of warming in the winter mean temperatures over parts of North America as evidence of a lack of warming of cold extremes (Cohen et al. 2021). We show that this is an incorrect interpretation.

We have added a reference to Fig 1 here and also added changed 'not representative' to 'not necessarily representative' (Line 292), to not overstate this point (as there are some regions where they are representative).

- line 351: please state the motivation to not remove the seasonal cycle. Is it to focus on the absolutely coldest, rather than relatively coldest, days?

Yes, it was to focus on the absolute coldest days, although we note the results and conclusions do not depend on this choice. This is now mentioned at Line 363.

References:

Blackport, R., J. A. Screen, K. van der Wiel, and R. Bintanja, 2019: Minimal influence of reduced Arctic sea ice on coincident cold winters in mid-latitudes. *Nat. Clim. Chang.*, **9**, 697–704, <https://doi.org/10.1038/s41558-019-0551-4>.

Cohen, J., K. Pfeiffer, and J. A. Francis, 2018: Warm Arctic episodes linked with increased frequency of extreme winter weather in the United States. *Nat Commun*, **9**, 869, <https://doi.org/10.1038/s41467-018-02992-9>.

Cohen, J., and Coauthors, 2020: Divergent consensus on Arctic amplification influence on midlatitude severe winter weather. *Nat. Clim. Chang.*, **10**, 20–29, <https://doi.org/10.1038/s41558-019-0662-y>.

Cohen, J., L. Agel, M. Barlow, C. I. Garfinkel, and I. White, 2021: Linking Arctic variability and change with extreme winter weather in the United States. *Science*, **373**, 1116–1121, <https://doi.org/10.1126/science.abi9167>.

Cohen, J., L. Agel, M. Barlow, C. I. Garfinkel, and I. White, 2022: Arctic change reduces risk of cold extremes—Response. *Science*, **375**, 729–730, <https://doi.org/10.1126/science.abn8954>.

Cohen, J., L. Agel, M. Barlow, and D. Entekhabi, 2023: No detectable trend in mid-latitude cold extremes during the recent period of Arctic amplification. *Commun Earth Environ*, **4**, 1–9, <https://doi.org/10.1038/s43247-023-01008-9>.

REVIEWERS' COMMENTS

Reviewer #2 (Remarks to the Author):

Third review of "Amplified warming of North American cold extremes linked to human-induced changes in temperature variability"

I thank the authors for their response to my prior review, and in particular their comprehensive discussion of the influence of the Cohen school of thought regarding cold extremes.

The editor has requested some specific comments on the results after applying the false discovery rate control. Like the authors, I am not overly concerned with the lack of gridbox-level significance of the trends, which is somewhat expected (especially after controlling for the mean or median), and not directly relevant for the detection and attribution results. However, the authors do need to correct their wording when describing the FDR control (and perhaps correct their methodology depending on what they are doing in their code). The relevant parameter for FDR control is the `alpha_FDR` on page 410 in the tracked changes version, which the authors set at 0.1. This value will be associated with a p-value cutoff, below which individual trends are found to be significant. So, the wording should be along the lines of "We control for a false discovery rate of 0.1". It is not clear what the authors mean by the stippling showing lack of significance at the 5% level in the context of FDR control, and this type of language should be removed.

After correction of this issue, I am happy to see the paper published.

We thank the reviewer again for their constructive comments which have helped improve the manuscript.

Reviewer #2 (Remarks to the Author):

Third review of "Amplified warming of North American cold extremes linked to human-induced changes in temperature variability"

I thank the authors for their response to my prior review, and in particular their comprehensive discussion of the influence of the Cohen school of thought regarding cold extremes.

The editor has requested some specific comments on the results after applying the false discovery rate control. Like the authors, I am not overly concerned with the lack of gridbox-level significance of the trends, which is somewhat expected (especially after controlling for the mean or median), and not directly relevant for the detection and attribution results. However, the authors do need to correct their wording when describing the FDR control (and perhaps correct their methodology depending on what they are doing in their code). The relevant parameter for FDR control is the `alpha_FDR` on page 410 in the tracked changes version, which the authors set at 0.1. This value will be associated with a p-value cutoff, below which individual trends are found to be significant. So, the wording should be along the lines of "We control for a false discovery rate of 0.1". It is not clear what the authors mean by the stippling showing lack of significance at the 5% level in the context of FDR control, and this type of language should be removed.

After correction of this issue, I am happy to see the paper published.

We agree that the wording of this was incorrect. We have changed the wording where this is brought up in the main text (L109), methods (L407) and in the Fig 1 and 3 figure captions as suggested by the reviewer.